# Eculizumab in C3 Glomerulopathy: A Systematic Review of Therapeutic Efficacy and Clinical Outcomes

**DOI:** 10.3390/pharmaceutics17101284

**Published:** 2025-10-01

**Authors:** Dominik Lewandowski, Mateusz Konieczny, Krzysztof Chrzanowski, Marta Jakubowska, Zuzanna Paryzek, Miłosz Miedziaszczyk, Ilona Idasiak-Piechocka

**Affiliations:** 1Students Research Group of Transplantation and Kidney Diseases, Poznan University of Medical Sciences, 60-352 Poznan, Poland; dlewandowski237@gmail.com (D.L.); konieczny.mateusz788@gmail.com (M.K.); chrzanowskik243@gmail.com (K.C.); marta.jakubowska.2001@gmail.com (M.J.); zuzanna.paryzek@gmail.com (Z.P.); 2Department of Clinical Pharmacy and Biopharmacy, Poznan University of Medical Sciences, 60-806 Poznan, Poland; 3Department of General and Transplant Surgery, Poznan University of Medical Sciences, 60-806 Poznan, Poland; ilonaidasiak@poczta.onet.pl

**Keywords:** C3 glomerulopathy, Eculizumab, dense deposit disease, C3 glomerulonephritis

## Abstract

**Background**: C3 glomerulopathies (C3G), including dense deposit disease (DDD) and C3 glomerulonephritis (C3GN), are rare kidney disorders driven by dysregulation of the alternative complement pathway. Eculizumab, a terminal complement inhibitor targeting C5, has emerged as a potential therapeutic option in these conditions. This systematic review evaluated the efficacy and safety of eculizumab in patients with C3G or DDD. **Methods**: Literature searches in PubMed and Cochrane databases identified case reports and case series reporting eculizumab use. **Results**: Only eight studies involving ten patients met the inclusion criteria. Eculizumab stabilized renal function and reduced proteinuria in most cases, especially when C5b-9 deposition was present. Histopathological improvements were variable, and recurrence after discontinuation occurred in some patients. Responses were limited in cases with alternative mechanisms of C5 activation. **Conclusions**: Eculizumab offers clinical benefit in select C3G and DDD patients but does not address the underlying cause of complement dysregulation. The need for long-term therapy, incomplete histologic resolution, and risk of relapse underscore the necessity of larger trials and the development of personalized treatment strategies.

## 1. Introduction

C3 glomerulopathies (C3G) are a rare group of kidney diseases in which the complement system is dysregulated in both the fluid phase and locally in the glomeruli. This results in a massive deposition of the C3 component in renal biopsies and low serum C3 levels. The two main disease entities within C3G, dense deposit disease (DDD) and C3-glomerulonephritis (C3GN) share common clinical and histopathological features. The disease tends to progress and relapse after kidney transplantation, and more than 50% of patients develop chronic renal failure [1,2,3].

The complement system is a cascade of enzymes belonging to innate immunity. It consists of soluble and membrane proteins that, when activated, generate potent effectors with inflammatory, cytotoxic, and immunomodulatory effects. It is involved in maintaining homeostasis and eliminating microorganisms [4,5]. Activation of the system via the alternative pathway leads to the formation of AP C3 convertase, production of anaphylatoxins (C3a, C5a), opsonization by C3b, formation of the membrane attack complex (MAC, C5b-9), and deposition of complement deposits in the kidneys. The kidneys are susceptible to such damage [3,6]. Chronic, uncontrolled activation of the alternative complement pathway is key in the pathogenesis of C3G. It can have an acquired or genetic basis. A common mechanism by which it is acquired is through autoantibodies directed against components of the C3 convertase and its regulators, which stabilize the enzyme complex and prolong its activity. These include antibodies against factors B, C3b, and H, as well as C3 nephritic factor (C3NeF), present in 40–80% of patients with C3G or immune-complex-dependent membranoproliferative glomerulonephritis (MPGN) [7]. C3NeF is an antibody found in all humans. Although the mechanisms underlying its formation are not fully understood, it is believed to result from an antigen-driven response leading to the production of high-affinity and high-specificity autoantibodies [8]. C3NeF is also associated with acquired partial lipodystrophy (APL), and as many as 25% of patients with this disorder develop C3G [9]. Among the hereditary mechanisms, mutations in genes encoding proteins involved in the alternative complement pathway, such as C3, factor H (CFH), and factor I (CFI), are distinguished. These mutations lead to the activation of the pathway. In one study, rare pathogenic variants in the CFH, CFI, or C3 genes were detected in 17% of patients with C3GN or Ig-MPGN; they were associated with factor H or I deficiency and worse renal prognosis regardless of age and treatment [10]. Additionally, gene copy number changes in the CFHR1–5 region, affecting factor H function, have been associated with both atypical hemolytic uremic syndrome (aHUS) and C3G, playing an important diagnostic and prognostic role [11]. In most cases of C3G, activation of the terminal complement pathway is also observed, as evidenced by the deposition of C5b-9 complexes. The presence of these deposits correlates with a worse prognosis—higher chronicity of biopsy lesions and a lower percentage of renal-event-free survival. Particularly high activity of the final pathway has been seen in cases with an immunofibroblastic signature [12]. The basic diagnostic tool remains a kidney biopsy with histological, immunofluorescent, and electron evaluation. The test shows the presence of C3 deposits in the absence of immunoglobulins, which distinguishes C3G from immune-complex-dependent MPGN [4]. Glomerular damage is chronic and leads to gradual loss of renal function [3].

The key role of the complement system in the pathogenesis of C3 glomerulopathy has become the basis for the development of targeted therapies for this system. An example of such a drug is eculizumab—a humanized monoclonal antibody with a high affinity for the complement protein C5, which inhibits the formation of proinflammatory fragments of C5a and the C5b-9 complex, acting as an inhibitor of the terminal phase of the complement (Figure 1). The efficacy and safety of eculizumab have been confirmed so far in the treatment of paroxysmal nocturnal hemoglobinuria (PNH) and atypical hemolytic uremic syndrome (aHUS), diseases in which the pathophysiology is based on excessive activation of the complement system [13,14].

## 2. Materials and Methods

### 2.1. Search Strategy

The preparation of this manuscript was based on the PRISMA Checklist. The authors used PubMed and Cochrane search engines where they inserted the following terms: “Eculizumab” AND “Glomerulopathy” OR “Nephropathy”. Case reports, clinical studies, randomized controlled trials, and non-randomized controlled trials were included. Due to the small number of studies, the year of publication did not serve as a criterion. The protocol of this systematic review was registered in the PROSPERO registry (CRD420251126078).

### 2.2. Inclusion Criteria

Inclusion criteria were established on the basis of the PICO framework. The population studied consisted of patients with C3 Glomerulopathy or Dense Deposit Disease (Population; P). The intervention involved treatment with Eculizumab (Intervention; I). The expected outcomes were to be compared to baseline values (Comparison; C). Final outcomes were assessed by comparing the initial and final glomerular filtration rate (GFR), creatinine level, and urine protein-to-creatinine ratio (UPCr) (Outcomes; O).

### 2.3. Exclusion Criteria

Exclusion criteria comprised conference abstracts, narrative reviews, cost-effectiveness analyses, letters to the editor, studies lacking scientific rigor, research not focused on C3G or not employing eculizumab, in vitro studies, studies that had not yet published their results, studies with follow-up periods longer than 36 months or shorter than 4 months, and research published in languages other than English. Due to the small number of studies, no other exclusion criteria were applied.

### 2.4. Data Collection Process

The selection process was conducted by four independently working researchers. All disagreements were discussed, and the consensus statement is provided in the text below. During the identification stage, 550 records were identified from the PubMed and Cochrane databases. After eliminating duplicates, 540 records were subjected to forward screening. A majority of these studies were excluded due to inadequate subject or form. Finally, 8 studies were involved in the current review. The search strategy included initial screening of titles and abstracts and subsequent full-text evaluation based on the predefined inclusion criteria. The data collection process is shown in Figure 2. in the form of a PRISMA flow diagram.

### 2.5. Bias Assessment

The risk of bias was evaluated using the Joanna Briggs Institute (JBI) Critical Appraisal Checklist for Case Reports [16] by two independently working researchers. Each case report was assessed using a written protocol. The JBI checklist comprises eight questions assessing methodological quality. A case report was classified as high quality only if it met at least seven criteria with affirmative (“Yes”) responses. Reports with more than one “No” or “Unclear” response were excluded from the final analysis. The results of the quality appraisal are presented in Section 3.

## 3. Results

### 3.1. Description of the Studies

This systematic review encompassed eight case reports and case series describing a total of ten patients treated with eculizumab for C3 glomerulopathy (C3G) or dense deposit disease (DDD). In the majority of cases, eculizumab therapy led to the stabilization of renal function and reduction in proteinuria, including improvements in the urine protein-to-creatinine ratio (UPCr). A case series by Le Quintrec et al. described three patients with C3G initially managed with mycophenolate mofetil, corticosteroids, and ACE inhibitors or angiotensin receptor blockers (ACEI/ARB). Two patients experienced clinical improvement following eculizumab treatment despite persistently low serum C3 levels and unchanged C3 deposits on repeat renal biopsy. Notably, one patient with recurrent C3G post-transplantation exhibited complete resolution of C3c deposits after therapy, indicating a histopathological response [17].

Two reports focused specifically on post-transplant disease recurrence. In the case presented by Sánchez-Moreno et al., a patient with recurrent DDD responded to eculizumab with a normalization of renal function and a marked reduction in UPCr (from 1.0 mg/mg to 0.1 mg/mg) following unsuccessful plasmapheresis [18]. Similarly, Gurkan et al. reported on a post-transplant patient initially treated with tacrolimus and corticosteroids. Renal function and UPCr improved significantly only after initiation of eculizumab. Although proteinuria recurred after nine months, ACEI therapy led to subsequent clinical stabilization [19]. Ozkaya et al. described a patient with nephrotic-range proteinuria (9.9 g/24 h) who experienced a substantial decrease to <0.2 g/24 h after eculizumab therapy. Previous treatment with corticosteroids and plasmapheresis had failed to control proteinuria [20]. A comparable outcome was observed in the case reported by Schmidt et al., where eculizumab monotherapy reduced C5b-9 and C3a levels, stabilized serum creatinine (from 2.5 to 1.5 mg/dL), and resulted in the resolution of albuminuria during follow-up [21]. The case described by Kasahara et al. involved a patient initially treated with corticosteroids and mizoribine. Following the introduction of eculizumab in combination with MMF, proteinuria decreased from 12.5 g/24 h to 2.5 g/24 h within one month, with renal function stabilizing over seven weeks. Despite significant improvement, the patient remained within the nephrotic range of proteinuria [22]. Two reports described less favorable outcomes. In the case reported by Leval et al., the patient did not respond to standard immunosuppressive therapy and experienced only transient improvement following eculizumab treatment. Clinical deterioration ensued within months. Nevertheless, follow-up renal biopsy revealed the disappearance of immune complexes, suggesting partial therapeutic efficacy [23]. Another case involving a likely C3G/aHUS overlap syndrome, complicated by recurrent pancreatitis, was reported by Jandal et al. The patient had been heavily pretreated with corticosteroids, MMF, cyclophosphamide, plasma exchange, and intravenous fluids. After eculizumab-induced partial remission lasting six months, the patient’s clinical condition gradually declined [24]. Table 1 presents the results obtained in the studies included in the systematic review. 

### 3.2. Statistical Analysis

Data are presented as medians (interquartile range). The Shapiro–Wilk test verified whether the parameter values followed a normal distribution. Student’s *t*-test compared differences between paired variables with a normal distribution. Wilcoxon’s test assessed differences between unpaired variables with non-normal distribution. All statistical analyses employed MedCalc, v. 23.1.7. (MedCalc Software Ltd., Ostend, Belgium). Treatment with eculizumab resulted in a non-significant reduction in the urine protein-to-creatinine ratio (UPCr), from a median of 438.78 mg/mmol (IQR: 113.00–1300.00) to 85.00 mg/mmol (IQR: 11.30–335.61) (*p* = 0.1064). Despite the downward trend, the difference did not reach statistical significance. In contrast, a statistically significant improvement was observed in estimated glomerular filtration rate (eGFR, CKD-EPI), which increased from a median of 44.39 mL/min/1.73 m*^2^* (IQR: 16.43–72.47) to 60.17 mL/min/1.73 m*^2^* (IQR: 47.64–101.86) (*p* = 0.0051). Similarly, serum creatinine levels decreased significantly, from a median of 1.87 mg/dL (IQR: 1.10–3.50) to 1.36 mg/dL (IQR: 0.80–1.70) (*p* = 0.0274). These findings suggest that while proteinuria reduction was not statistically significant, eculizumab treatment was associated with measurable and significant improvements in renal function.

### 3.3. Bias Assessment

The Bias Assessment of the 13 included studies was evaluated using the Joanna Briggs Institute (JBI) Critical Appraisal Checklist for Case Reports, which consists of eight domains assessed with responses of “Yes”, “No”, or “Unclear” (Table 2). The domains are as follows: D1: Were the patient’s demographic characteristics clearly described? D2: Was the patient’s history clearly described and presented in a timeline? D3: Was the clinical condition at presentation clearly described? D4: Were diagnostic tests and results clearly described? D5: Were the interventions or treatment procedures clearly described? D6: Was the post-intervention clinical condition clearly described? D7: Were adverse or unanticipated events identified and described? D8: Did the case report provide a clear takeaway message? Three studies received affirmative responses across all domains and were classified as high quality [17,19,20]. Conversely, four studies were rated as low quality and excluded from the outcome analysis [25,26,27,28]. The case report by Besbas et al. failed to provide sufficient clinical context and omitted a clear description of the patient’s medical history. Moreover, it did not address adverse or unanticipated events (D7), nor did it include a meaningful takeaway message (D8), significantly limiting its value for inclusion [25]. Similarly, the reports by Parra et al. and Bomback et al. involved cohorts in which only a subset of patients were diagnosed with C3G or DDD. Critical data were missing, including individual patient characteristics, clinical histories, post-treatment outcomes, and adverse events—precluding a reliable evaluation of these cases within the scope of this review [26,27]. The study by Ruiz-Fuentes et al., describing a post-transplant patient with C3G in the context of chronic kidney disease and monoclonal gammopathy, was also excluded. The report lacked essential information regarding the patient’s post-intervention status and any adverse effects observed, resulting in two negative domain assessments and a classification of insufficient quality [28]. Three additional studies met all inclusion criteria except for the documentation of adverse or unexpected events (D7). Given the relevance of safety reporting—particularly when investigating a novel therapeutic agent—this omission was noted. However, as all other quality domains were fulfilled, these studies were acceptable for inclusion in the final analysis [18,21,24]. The case report described by Kasahara et al. provided all necessary information; however, it failed to present the patient’s history in a timeline [22]. Similarly, a study performed by Levart et al. failed to describe the post-intervention clinical condition of the patient; however, this was marked as a “high quality” paper, as that was the only domain with a negative answer [23].

## 4. Discussion

Eculizumab exerts its effects by inhibiting the complement system at the level of the C5 protein. The changes characteristic of C3G and DDD are primarily associated with the dysfunction of the alternative complement pathway at the C3 level and the deposition of C3 fragments in the kidneys. The common denominator between the action of eculizumab and the pathophysiology of C3G and DDD is the involvement of the C3 and C5 proteins within the same alternative complement activation pathway. Although eculizumab’s mechanism of action does not directly affect C3, this drug effectively inhibits the generation of the proinflammatory C5a fragment and the membrane attack complex (MAC, C5b-9). As an inhibitor of terminal complement activation, it limits renal damage associated with the formation of MAC. However, the effect of eculizumab on C3 deposition remains unclear. In the case series described by Quintrec et al., both clinical improvement without changes in C3 deposits and their complete resolution following therapy were observed [17]. The histopathological effects of anti-C5 therapy, therefore, require further verification in studies with larger sample sizes.

Our results indicate that inhibition of MAC formation alone is sufficient to achieve a significant increase in eGFR and a decrease in UPCr. It appears that the key factor in eculizumab’s efficacy is the presence of C5b-9, which the drug can directly affect. Therefore, the detection of MAC in renal biopsy material could serve as a predictor of a good response to therapy [27]. Similarly presented case reports show that there is a higher possibility of response to eculizumab treatment in patients with recurrent C3G or DDD after kidney transplantation, however more studies on this subgroup of patients are needed to fully assess the drug’s efficacy [18,19]

A limited efficacy of anti-C5 treatment could be observed in patients with C3NeF autoantibodies, in whom C3 activation and deposition were likely the main drivers of renal damage [29]. In such cases, factor B or D inhibitors appear to be more clinically appropriate; however, there are currently no comparative studies assessing the efficacy of these drugs in the context of C3G and DDD.

Another limitation of eculizumab therapy is the presence of irreversible structural changes in the kidneys, such as fibrosis or sclerosis, which result from chronic damage. This suggests that as the disease progresses and irreversible changes intensify, the response to treatment may gradually decline [30]. Furthermore, complement activation can occur locally, independently of circulation. The kidneys are capable of locally synthesizing complement components in response to inflammatory stimuli—both acute and chronic [31]. This phenomenon may limit the availability of eculizumab at the site of activation and, consequently, reduce its therapeutic efficacy. In some cases, C5 activation may occur independently of the alternative pathway, such as through the action of thrombin or trypsin. It is worth noticing that there is evidence of trypsin expression in the epithelial cells of the kidneys, as well as in other tissues such as the skin, esophagus, stomach, small intestine, lungs, liver, extrahepatic bile ducts, and also in cells of the spleen and neurons. This pathway bypasses eculizumab’s mechanism of action, limiting its therapeutic efficacy [32,33,34]. Additionally it has been demonstrated that neutrophil-derived neutrophil elastase (HNE) activates the alternative complement pathway by cleaving C5 in the presence of C6 [35]. Furthermore, there is a possibility that genetic factors contribute to the drug’s lack of efficacy—an example is the p.Arg885His mutation, which prevents eculizumab from binding to C5 [36]. A summary of the predictive factors for eculizumab treatment is presented in Table 3. Cases of hereditary, complete, or partial complement control proteins, such as CD55 deficiency, have been described and may result in a dysregulation of complement activity. However, the available data do not conclusively support the notion that this deficiency directly predisposes a patient to the development of C3G or DDD. Similarly, there is no evidence that mutations in the gene encoding CD59 are an etiologic factor in these diseases.

Currently, no universally accepted or specific treatment for C3G exists. The interpretation of clinical trials is challenging due to small sample sizes and disease heterogeneity. Nevertheless, there is evidence supporting the efficacy of both classical therapeutic approaches—such as immunosuppressive agents, plasma infusions, and plasmapheresis—and emerging therapies targeting components of the complement cascade, including inhibitors of C3 and C5 convertases.

Angiotensin-converting enzyme inhibitors or angiotensin-receptor blockers should be considered as a first line of treatment for proteinuria associated with glomerulopathy, as such treatment leads to better renal survival than the use of immunosuppressive agents [37]. There is, however, a report showing better outcomes in patients suffering from DDD who underwent therapy with the use of both renin angiotensin blockers and immunosuppressive agents [38]. An additional benefit for C3G patients may be obtained through the use of statins; however, the majority of examined patients also suffered from dyslipidemia, so it is difficult to state whether statins have independent nephroprotective effects or the effect of correcting the lipid levels [39]. Immunosuppressive agents are a disputable method of treatment. The main example is rituximab which, despite its efficacy in the treatment of Membranoproliferative glomerulonephritis (MPGN), shows no efficacy in the treatment of C3G and DDD [40]. Data on the use of immunosuppressive agents is limited, and large prospective studies are needed; however, the current standard of care includes the use of Mycophenolate Mofetil (MMF) and corticosteroids. There are several studies showing the effectiveness of MMF in C3G. A retrospective study, performed in 35 Spanish hospitals and including 97 patients with C3G and DDD, showed that 79% of the patients receiving MMF and corticosteroids achieved partial or complete remission. The same study showed that only 26% of patients receiving other immunosuppressants achieved remission. Use of MMF and corticosteroids also made the risk of kidney failure, described as eGFR lower than 15, equal to only 14% in comparison to a 60% risk when different immunosuppressants were used [41]. Another study examining 60 patients suffering from C3G proved that no patient treated with MMF and corticosteroids developed end-stage renal disease compared to 10 patients treated with corticosteroids alone or corticosteroids with cyclophosphamide [42]. Despite these described outcomes, there is proof that heavier proteinuria and lower soluble-membrane-attack-complex levels at the beginning of therapy are associated with resistance to MMF, and even in patients responding to treatment, remission rates equal 67%, which is a significantly lower number than in previously described studies, but this drug still outperforms other immunosuppressive agents [43].

There is very scarce data considering the use of plasma therapy in C3G, what exists is limited to only case reports and case series. One of the problems associated with C3G treatment is the lack of successful therapeutic options when recurrence occurs. Two studies describing five patients show that multimodal treatment including plasmapheresis and immunosuppressive agents can delay the onset of renal failure. One possible explanation is the removal of C3NeF from blood; however, measured C3NeF did not correlate with disease course [44,45]. Taking into consideration that plasmapheresis was only one part of a complex treatment plan, and the fact that data is significantly limited, this way of treatment should be considered only as a last resort, and studies should focus on searching for more specific options. The lack of efficacy of therapies such as plasmapheresis or rituximab may be due to the fact that removing circulating antibodies does not significantly affect their de novo synthesis. Since autoantibodies rapidly regenerate, this diminishes any potential therapeutic effect. Furthermore, it should be emphasized that in most patients with C3G and DDD, the dominant pathological mechanism is chronic, uncontrolled activation of the alternative complement pathway, not the presence of autoantibodies per se. Therefore, the efficacy of therapies targeting B lymphocytes remains equivocal and requires further study [40]. It is important to emphasize that not all patients with these conditions are positive for C3NeF. This autoantibody was detected in approximately 86% of patients with DDD and in 49% of patients with MPGN type I or C3GN, further emphasizing the heterogeneity of the immunopathogenesis of these conditions [37]. Another therapeutic option is kidney transplantation, but there is only one retrospective study from a single institution, which included 21 patients. The study showed a high rate of recurrence after transplantation—66.7%—occurring in 2–3 years’ time, and half of those patients experienced graft failure [46]. The main risk factors for recurrence are delayed graft function, infection, and neoplasia. Despite several ongoing studies aiming to assess management of C3G in kidney transplant recipients, there are no guidelines on how to treat this condition [47].

As described in this unit, nonspecific therapies have their limitations, however, nowadays there is a visible shift to develop novel therapeutic agents directed at the complement system, as these are expected to have the best results. Pegcetacoplan is a targeted C3 and C3b inhibitor mitigating complement-mediated kidney damage. The study evaluating the drug’s efficacy in a C3G cohort and proved that pegcetacoplan reduced mean proteinuria by 50,9% and stabilized mean serum albumin and eGFR over 48 weeks of study length [48]. Another drug with similar mechanisms of action, aiming to prevent C3 deposition and renal inflammation is Iptacopan, which was evaluated in a study performed in 2023. It showed that urine protein to creatinine ratio (UPCR) decreased by 45%, median C3 deposit score decreased by 2.5 (scale 0–12) and serum C3 levels normalized in the majority of patients [49]. The last drug from the C3/C3 Convertase inhibitors group is danicopan, however there is only one study available, assessing correlations between complement biomarkers, clinical parameters, and biopsy scores, and there is further research needed to precisely verify the patient’s responsiveness to the drug [50]. All three described drugs are in phase 2 of development, so despite favorable outcomes and safety profiles being assessed, there is a need to continue clinical trials to implement them into clinical practice. The other group of novel therapeutic agents are C5/C5-Convertase inhibitors including eculizumab, ravalizumab, and avacopan. There are only two clinical studies regarding use of avacopan in C3G and their results are inconsistent. A study performed in 2022 compared the C3G Histological Index (C3GHI) in patients treated with avacopan or placebo, and it showed a beneficial effect of avacopan [51]; however, a study performed in 2025 assessing the safety and efficacy of the drug did not met the primary endpoint and the clinical effects were too variable to draw significant conclusions [52]. The last new therapeutic option is ravulizumab, being a modified version of eculizumab with a longer half-life that shows efficacy in the treatment of aHUS and potentially may be used in the management of C3G [53]. Future clinical studies would greatly contribute to this field by comparing the safety and efficacy of eculizumab and the described novel therapeutic agents.

Eculizumab should be considered a targeted therapy that acts on the pathogenetic mechanism directly responsible for kidney damage in C3G and DDD. Current research focuses on the causal treatment of glomerulopathy, preventing secondary, irreversible damage. However, it should be emphasized that although eculizumab does not address the primary cause of complement activation in C3G and DDD, this systematic review supports its clinical efficacy.

Anti-C5 therapy appears to require chronic and potentially indefinite use to maintain its therapeutic effect. To date, the possibility of achieving sustained remission after treatment has not been confirmed, and there is a lack of data indicating that this drug can be safely discontinued. Eculizumab remains a relatively novel therapeutic option in the treatment of C3G and DDD, which is associated with numerous research gaps and unresolved clinical issues. In majority of assessed studies, researchers did not determine the levels of antibodies in patients receiving eculizumab treatment—studies examining this dependence can possibly explain insufficient answer to this drug in certain patients. Further randomized controlled trials in larger patient populations are necessary to confirm its efficacy and safety, considering the previously mentioned predictive factors, comparisons with other drugs, and potential combination therapies. Improving our understanding of this area may enable the development of personalized and; therefore, more effective therapeutic strategies.

## 5. Limitations

This systematic review is subject to several important limitations. First and foremost, all included studies were individual case reports or small case series, lacking control groups or randomization. This inherently limits the strength of the evidence. The absence of comparative arms makes it impossible to differentiate between spontaneous improvement, placebo effect, and true treatment efficacy. Second, the overall number of eligible studies was very limited. Only eight publications describing ten patients met the inclusion criteria, which severely restricts the generalizability of the findings. The small sample size also limits the ability to draw conclusions—particularly regarding changes in key clinical endpoints such as the urine protein-to-creatinine ratio (UPCr), where observed reductions were promising but not statistically significant. Additionally, this review was limited to studies published in English. Furthermore, variability in diagnostic methods, treatment duration, follow-up length, and reporting quality across case reports makes standardized comparison challenging.

## 6. Conclusions

Eculizumab demonstrates therapeutic potential in C3 glomerulopathy and dense deposit disease by targeting terminal complement activation. It can stabilize kidney function and reduce proteinuria in select patients, particularly those with evidence of C5b-9 deposition. However, the response to therapy is heterogeneous and may be limited by chronic structural damage, alternative complement activation pathways, or genetic mutations affecting C5 binding. As eculizumab does not correct the underlying mechanism of complement dysregulation, sustained therapy may be required to maintain clinical benefit. The main limitation of this study is the fact that only 10 patients were involved in the final analysis due to the frequency of the disease and difficulties in creating bigger clinical studies. Further large-scale, controlled studies are essential to better define its role, identify predictive markers of response, and explore combination or alternative-complement-targeted therapies. A head-to-head study assessing the efficacy and safety of eculizumab vs. iptacopan/pegcetacopan seems necessary.

## Figures and Tables

**Figure 1 pharmaceutics-17-01284-f001:**
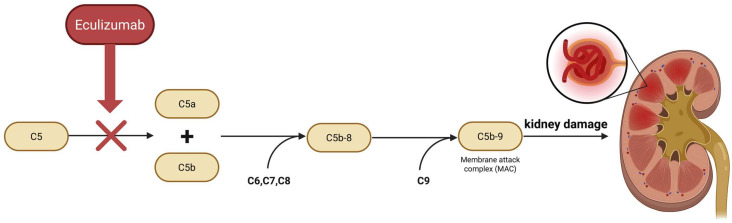
Mechanism of action of Eculizumab [15]. Created in BioRender. Konieczny, M. (2025) https://BioRender.com/364tr9n (accessed on 30.09.2025).

**Figure 2 pharmaceutics-17-01284-f002:**
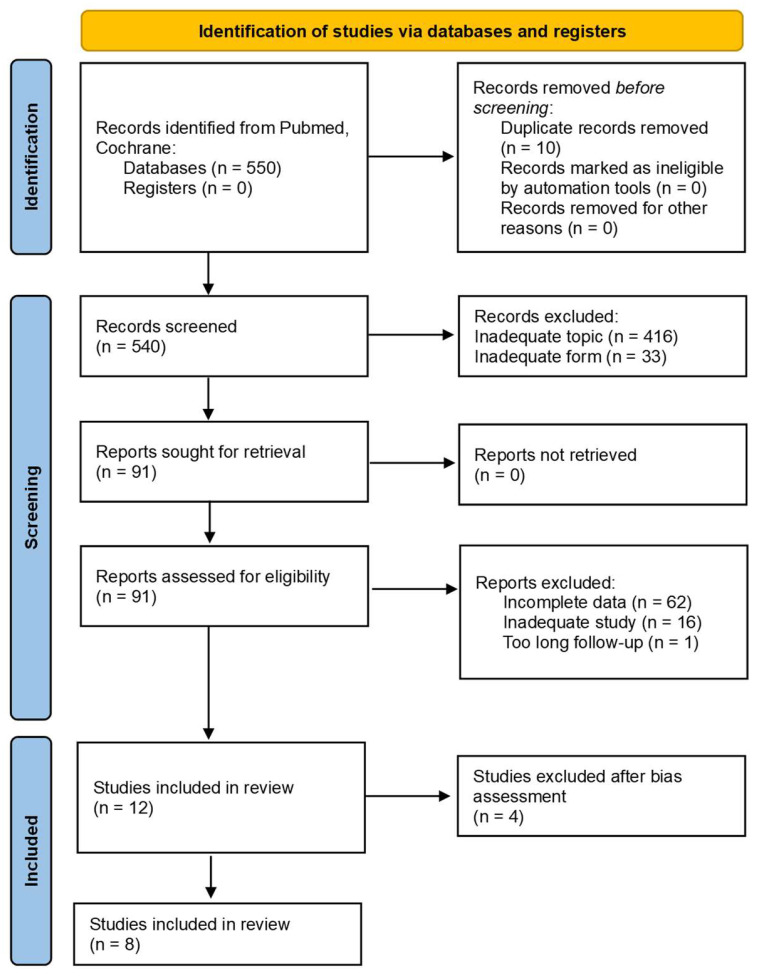
Data collection process—PRISMA flow diagram.

**Table 1 pharmaceutics-17-01284-t001:** Studies’ characteristics.

Study	Illness	Initial GFR	Final GFR	Initial Creatinine (mg/dL)	Final Creatinine (mg/dL)	Observation Time (Months)	Initial UPCr (g/g)	Final UPCr (g/g)
Le Quintrec et al. [17]	C3G	8.87	33.01	6	2	19	12.55	0.71
	C3G	23.17	67.31	2.2	0.9	32	1.41	0.8
	C3G	16.43	47.64	4.1	1.7	6	11.49	6.9
Ozkaya et al. [20]	DDD	184.28	180.03	0.49	0.5	7	-	-
Sanchez-Moreno et al. [18]	DDD	72.47	101.86	1.1	0.8	30	1	0.1
Kasahara et al. [22]	DDD	80.27	107.76	1.02	0.78	12	-	-
Levart et al. [23]	C3G	65.82	53.03	1.54	1.82	24	-	-
Jandal et al. [24]	C3G	15.98	47.96	3.5	1.41	6	6.36	2.97
Schmidt et al. [21]	C3G	23.01	42.66	2.5	1.5	5	-	-
Gurkan et al. [19]	C3G	65.61	77.45	1.5	1.3	12	3	1.3

**Table 2 pharmaceutics-17-01284-t002:** Bias Assessment.

Study	D1	D2	D3	D4	D5	D6	D7	D8
Sanchez Moreno et al. [18]	Yes	Yes	Yes	Yes	Yes	Yes	No	Yes
Le Quintrec et al. [17]	Yes	Yes	Yes	Yes	Yes	Yes	Yes	Yes
Ozkaya et al. [20]	Yes	Yes	Yes	Yes	Yes	Yes	Yes	Yes
Gurkan et al. [19]	Yes	Yes	Yes	Yes	Yes	Yes	Yes	Yes
Besbas et al. [25]	Unclear	No	Yes	Yes	Yes	Yes	No	No
Parra et al. [26]	No	No	Yes	Yes	Yes	No	No	Yes
Bomback et al. [27]	No	No	Yes	Yes	Yes	No	No	Yes
Ruiz-Fuentes et al. [28]	Yes	Yes	Yes	Yes	Yes	No	No	Yes
Schmidt et al. [21]	Yes	Yes	Yes	Yes	Yes	Yes	No	Yes
Jandal et al. [24]	Yes	Yes	Yes	Yes	Yes	Yes	No	Yes
Kasahara et al. [22]	Yes	No	Yes	Yes	Yes	Yes	Yes	Yes
Levart et al. [23]	Yes	Yes	Yes	Yes	Yes	No	Yes	Yes

Judegement: Yellow—some concerns, Green—Low.

**Table 3 pharmaceutics-17-01284-t003:** Predictive factors of eculizumab treatment response.

Factor	Better/Worse Response to Treatment	Explanation
C5b-9/MAC deposition on renal biopsy	Better response	Eculizumab blocks C5, therefore preventing formation of C5b-9. Cases with intrarenal C5b-9 showed clinical improvement after therapy [17,21]
Minimal fibrosis/sclerosis on renal biopsy	Better response	Eculizumab cannot reverse structural damage. Patients without chronic, irreversible changes more often show recovery [1,37]
Elevated soluble C5b-9	Better response	Eculizumab reduces C5b-9 and C3a accordingly with clinical improvement [21]
Post-transplant recurrent disease	Better response	Several post-transplant patients with recurrent C3G/DDD improved after eculizumab treatment [18,19]
Presence of C3NeF	Worse response	When dominant pathogenic mechanism of complement activation is C3 deposition driven by C3NeF, there is smaller impact of blocking C5 [37]
Genetic C5 variants preventing eculizumab binding	Worse response	Certain C5 polymorphisms block eculizumab binding (e.g., p.Arg885His) [34]
Advanced chronic kidney disease	Worse response	Longstanding disease with following fibrosis or sclerosis gives limited chance for recovery [38]
Overlap syndromes	Worse response	Overlap syndromes may alter the mechanism of injury [24]

## Data Availability

All new data are available in article.

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
