# Peer review of "Eculizumab in C3 Glomerulopathy: A Systematic Review of Therapeutic Efficacy and Clinical Outcomes"

_pharmaceutics, 2025, doi:10.3390/pharmaceutics17101284_

Round 1
Reviewer 1 Report
Comments and Suggestions for Authors
The manuscript by Lewandowski et al. is devoted to the analysis of literature data on the efficacy of eculizumab in the treatment of C3 dense deposit disease (DDD) and C3 -glomerulonephritis. The task of the review is quite complex, since despite the large number of publications, there is little reliable data on this topic. As a result, they analyzed 10 cases. The overall conclusion is that eculizumab has a positive effect, but information on the efficacy of long-term treatment, the risk of relapses and histological data is insufficient. The article is useful and can be published in the journal Pharmaceutics. I have some minor comments.
Line 52 MPGN needs to be decrypted
In Table 1 it is worth indicating changes in the C3 level for individual patients, changes in UPCr, units of creatinine and time units.
Was the determination of antibodies to Ecu carried out? Can this factor be excluded to explain the insufficient effectiveness of Ecu?
Author Response
The authors thank you very much for the effort put into the review and for valuable comments. The comments helped us significantly improve the article.
Reviewer 2 Report
Comments and Suggestions for Authors
pharmaceutics-3857727
Title: Eculizumab in C3 Glomerulopathy: A Systematic Review of Therapeutic Efficacy and Clinical Outcomes
Authors: Lewandowski D., et al.
This review analyzes the use of the monoclonal antibody Eculizumab’s within the clinic to treat a rather rare form of kidney diseases, the C3 glomerulopathies. On the whole the review is well written and thought provoking.
When discussing potential alternative mechanisms of C5 activation (lines 250-258) the Authors mention trypsin. To my knowledge, expression of this enzyme is restricted to the pancreas and it is only found within the digestive tract. What is the evidence that it is found in kidney? However, there are a number of serine proteases within the coagulation cascade, as well as proteases released by immune cells, e.g. neutrophil elastase, that better fit into the proposed concept, which should be discussed.
Additionally, I am missing a discussion on the role of complement control proteins, such as CD55 and CD59. Have deficiencies in these factors been identified as contributing factors to C3G and DDD?
The autoantibody C3NeF is considered a contributing factor in patients with C3G and DDD. How does this arise? Plasmapheresis and rituximab have been successful in treating other forms of autoantibody-induced kidney disease (e.g. membranous nephropathy). I find it surprising that these are ineffective here. Is it possible that only a subset of patients possess C3NeF?
At the beginning of the Results section (line 127/128), the Authors state that only 10 patients fulfilled their requirements. I find this number quite surprisingly small, as the clinical studies performed to test the safety and effectiveness of these treatments must have involved more patients. Please comment.
Minor points
line 52 define MPGN
line 93 use ‘small number’ not ‘little amount’
line 104/105 see comment above
line 299/300 C3NF is this the same as C3NeF (line 51)? If so, please fix
Author Response

(The authors gave the same response as above.)

Reviewer 3 Report
Comments and Suggestions for Authors
Lewandowski et al. have conducted a systematic review on the use of eculizumab in C3G and DDD, evaluating the therapeutic efficacy, clinical outcomes, and safety. Within a database of 550 records, only 8 case reports covering 10 patients were included. The reports indicated that eculizumab has the potential to stabilize renal function and reduce proteinuria, improve eGFR and creatinine level, although the histological response and durability can be inconsistent. Lewandowski et al. concluded from these limited reports that eculizumab provides benefit in certain patient populations and call for further exploration of complement inhibitors in patients. However, there are the following issues that the authors should address before the acceptance of this review article.
Major:
- This review work solely relies on case reports with a very small sample size, hence, the generalizability is limited. The authors should more explicitly emphasize this limitation in the abstract and conclusion.
- Please be more specific about the details of search strings and inclusion/exclusion criteria for the PRISMA method, since the current presentation is too vague.
- The readers could benefit from learning more clinical implications with this review work, for example, the authors already mentioned that those with C5b-9 deposition tend to respond better, a summary table across other predictors would add depth to this research, such as other autoantibodies or presentation of fibrosis. On the other hand, who are the patients that are unlikely to respond, a discussion around non-responders and their predictor biomarkers would be appreciated. This could add more impact to this work rather than a summary of a small sample size of case reports.
- While the discussion touched on newer complement inhibitors, the core section or the article itself seems narrowly focused on eculizumab outcomes, as we have newer agents available, the clinical implications from an eculizumab-only study are questionable.
Minor:
- Please be consistent in the reference list, right now they are ending with a mix of URL and DOI. Please ensure all of them match the journal guidelines.
Author Response

(The authors gave the same response as above.)

Round 2
Reviewer 2 Report
Comments and Suggestions for Authors
The Authors have addressed my concerns.
Reviewer 3 Report
Comments and Suggestions for Authors
The authors have addressed my comments, no further issues.